# Metformin protects trabecular meshwork against oxidative injury via activating integrin/ROCK signals

**Lijuan Xu, Xinyao Zhang, Yin Zhao, Xiaorui Gang, Tao Zhou, Jialing Han, Yang Cao, Binyan Qi, Shuning Song, Xiaojie Wang, Yuanbo Liang***

State Key Laboratory of Ophthalmology, Optometry and Visual Science, National Clinical Research Center for Ocular Diseases, Eye Hospital, Wenzhou Medical University, Wenzhou, Zhejiang, China

**Abstract** This study aimed to investigate the protective effect of metformin on trabecular meshwork (TM) and explore its molecular mechanisms in vivo and in vitro. Ocular hypertension (OHT) mouse models were induced with dexamethasone and further treated with metformin to determine the intraocular pressure (IOP)-lowering effect. Cultured human TM cells (HTMCs) were prestimulated with tert-butyl hydroperoxide (tBHP) to induce oxidative damage and then supplemented with metformin for another 24 hr. The expression of fibrotic markers and integrin/Rho-associated kinase (ROCK) signals, including α-smooth muscle actin (α-SMA), transforming growth factor-β (TGF-β), fibronectin, integrin beta 1, ROCK 1/2, AMP-activated protein kinase, myosin light chain 1, and F-actin were determined by western blotting and immunofluorescence. Reactive oxygen species (ROS) content was analysed using flow cytometry. Transmission electron microscopy was performed to observe microfilaments in HTMCs. It showed that metformin administration reduced the elevated IOP and alleviated the fibrotic activity of aqueous humour outflow in OHT models. Additionally, metformin rearranged the disordered cytoskeleton in the TM both in vivo and in vitro and significantly inhibited ROS production and activated integrin/ROCK signalling induced by tBHP in HTMCs. These results indicated that metformin reduced the elevated IOP in steroid-induced OHT mouse models and exerted its protective effects against oxidative injury by regulating cytoskeleton remodelling through the integrin/ROCK pathway. This study provides new insights into metformin use and preclinical evidence for the potential treatment of primary open-angle glaucoma.

**\*For correspondence:**
yuanboliang@126.com

**Competing interest:** The authors declare that no competing interests exist.

## Editor's evaluation

The claims that metformin reduced elevated intraocular pressure in vivo and regulated cytoskeleton remodeling in vitro are supported by convincing data. This study provides a new direction for further exploration toward the treatment of primary open-angle glaucoma.

## Introduction

Intraocular pressure (IOP) elevation, predominantly resulting from increased resistance to aqueous humour outflow (AHO), is a major risk factor for primary open-angle glaucoma (OAG) deterioration (*Casson et al., 2012*; *Wu et al., 2020*). The only proven method is IOP lowering (*Richter and Coleman, 2016*). According to Bill and his colleagues (*Bill and Hellsing, 1965*; *Bill and Svedbergh, 1972*), up to 80% aqueous humour is drained via conventional trabecular meshwork (TM) pathway; however, available anti-glaucoma medications mostly act on sites other than TM and have limited

efficiency. Therefore, polypharmacy has become increasingly prevalent, coupled with an increasing economic burden on society and patients (*Wu et al., 2020*).

IOP elevation is associated with TM stiffness (*Alvarado et al., 1984*; *Heijl et al., 2002*; *Johnstone et al., 2021*). Theoretically, drugs promoting the recovery of damaged TM are potentially effective in lowering IOP. Integrin and Rho-associated protein kinase (ROCK) play pivotal roles in cytoskeletal formation and maintenance (*Tan et al., 2020*; *Yemanyi et al., 2020*), and ROCK inhibitors (ROCKi) can decrease actomyosin contraction and actin crosslinking (*Liu et al., 2021*). ROCKi are the only drug that directly target conventional outflow function (*Aga et al., 2008*; *Rao et al., 2001*; *Ren et al., 2016*). It alters the architecture of AHO and expands the juxtacanalicular connective tissue region. Currently, the clinically available ROCKi include Y-27632 and ripasudil.

Metformin (MET), an oral biguanide insulin-sensitising drug, is the most widely used treatment for type 2 diabetes mellitus (DM) (*Foretz et al., 2014*). It is a multifunctional drug (*Rangarajan et al., 2018*; *Zhao et al., 2021*). Studies by *Lin et al., 2015* and *Maleškić et al., 2017* found that MET reduced the risk of OAG in patients with DM, and this effect persisted even after controlling for glycated haemoglobin. However, this effect was not observed with other hypoglycaemic medications (insulin, sulfonylureas, thiazolidinediones, and meglitinides), suggesting that the protective effect of MET on glaucoma goes beyond glycaemic improvement. However, the precise mechanisms involved remain unclear.

Excessive reactive oxygen species (ROS) in TM play an important role in the disruption of cytoskeletal integrity and apoptosis (*Hu et al., 2017*), leading to pathological alterations in AHO and subsequent IOP rise (*Babizhayev and Bunin, 1989*; *Saccà et al., 2016*). Reduction of intracellular ROS levels by MET via activating AMPK signal has been reported in primary hepatocytes (*Ota et al., 2009*), vestibular cells (*Lee et al., 2014*), and human immune cells (CD14+ monocytes, CD3+ T cells, CD19+

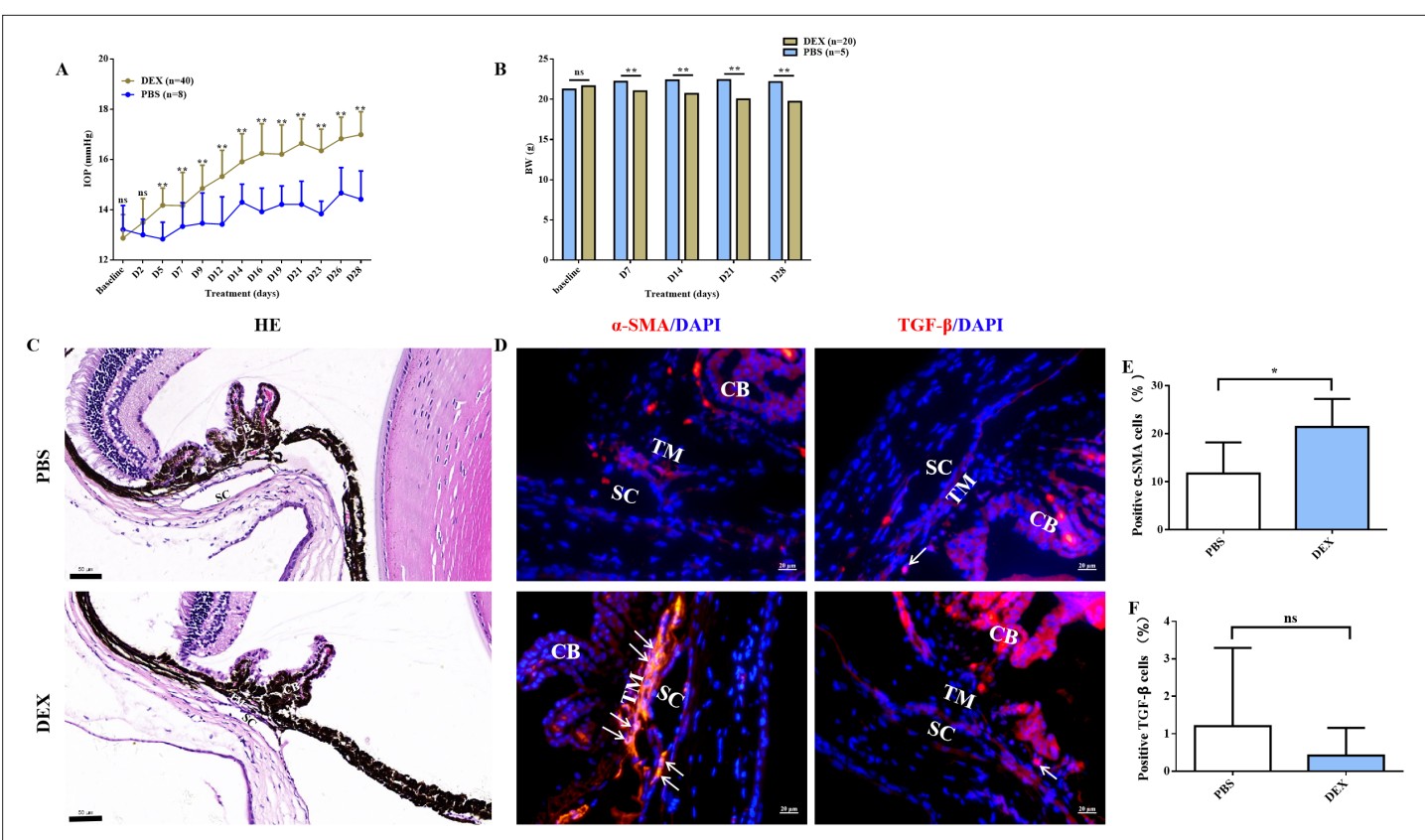

**Figure 1.** Topical ocular DEX-induced OHT in mice. (**A**) Elevated IOP in DEX-treated C57BL/6J mice was induced significantly at 4 weeks (p < 0.01). (**B**) The body weight comparison between the DEX-treated mice and control. (**C**) HE staining of OHT models. Scale bar, 50 μm. (**D**) α-SMA and TGF-β staining in the representative OHT models. Scale bar, 20 μm, (**E, F**) Quantification of α-SMA and TGF-β of the models. *p < 0.05, **p < 0.01, ns: non-significance, DEX: dexamethasone, PBS: phosphate-buffered saline, HE: haematoxylin and eosin, IOP: intraocular pressure, OHT: ocular hypertension, BW: body weight, TM: trabecular meshwork, SC: Schlemm's canal, CB: ciliary body, α-SMA: α-smooth muscle actin, TGF-β: transforming growth factor-β.

B cells, and CD56⁺ NK cells) (**Hartwig et al., 2021**). Conversely, there is also evidence on cellular ROS level increase in some cancer cells after MET treatment, including AsPC-1 pancreatic (**Warkad et al., 2021**), osteosarcoma (**Li et al., 2020a**), and breast cancer cells (**Yang et al., 2021**). These seemingly contradictory results suggested that MET plays different roles under different metabolic environments. To investigate the role of MET in damaged TM cells and ocular hypertension (OHT) mouse models, we used tert-butyl hydroperoxide (tBHP) to induce oxidative damage in TM cells (**Tang et al., 2013**; **Wang et al., 2021**) and topical glucocorticoids to create OHT mouse models (**Li et al., 2021**). The results showed that MET protected against cytoskeletal destruction in TM by activating the integrin/ROCK pathway and alleviated elevated IOP in steroid-induced OHT mouse models.

## Results

### Steroid-induced OHT in mice

The experiment included two steps: first, OHT modelling and the subsequent drug's IOP-lowering effect test were performed, both of which lasted for 28 days. A steroid-induced OHT mouse model was successfully established bilaterally in the first step. The baseline IOP did not differ between dexamethasone (DEX)-treated and phosphate-buffered saline (PBS)-treated vehicle eyes; however, starting at day 5, IOP was significantly elevated in DEX-treated eyes ($p < 0.01$, **Figure 1A**) and stabilised by day 28 (defined as an increase by >3 mm Hg or 30% from the baseline in at least two of the three IOP tests [days 23, 26, and 28]). The body weight (BW) was significantly lower in DEX-treated mice than in controls from days 7 to 28 (**Figure 1B**). Approximately 90.0% DEX-treated eyes had elevated IOP after 28 days of steroid induction with an average increased magnitude of 4.12 ± 1.26 mm Hg (2.00–7.30 mm Hg). Moreover, the fibrotic marker (α-SMA) was obviously overexpressed in the OHT mouse models ($p < 0.05$, **Figure 1D–F**).

### MET effectively reversed steroid-induced OHT in mice

In this step, successfully DEX-induced OHT mouse models were randomly assigned into three groups with the continued prescription of DEX: DEX withdrawal (group 2, $n = 10$), DEX + PBS (group 3, $n = 8$), and DEX + 0.3% MET (group 4, $n = 18$). Meanwhile, a group of mice were treated with solo PBS as control with a duration of 8 weeks (group 1, $n = 8$).

The experimental procedure is illustrated in **Figure 2A**. The IOPs after 28 days of steroid induction in all three OHT groups were similar (17.27 ± 1.08, 16.91 ± 1.01, and 17.06 ± 0.82 mm Hg, respectively). Compared with group 3, DEX-withdrawal mice had significantly reduced IOP after 17–28 days (all $p < 0.05$). After 5 days of MET treatment, IOPs significantly reduced in group 4 ($p < 0.01$, **Figure 2B**). In fact, MET almost completely neutralised steroid-induced OHT, returning IOP to near DEX-withdrawal levels on day 21, suggesting a therapeutic role of MET in OAG. Specific IOP values are shown in **Figure 2B**.

### MET attenuated the steroid-induced TM cytoskeletal destruction in vivo

As shown in **Figure 3**, MET improved fibrosis and the intensity of phalloidin labelling of F-actin in the TM tissue of steroid-induced OHT C57BL/6 mice. Quantitative comparison showed a significant difference in the number of α-SMA- and fibronectin (FN)-positive TM cells between MET-treated and control mice (**Figure 3C, E**). Additionally, MET treatment promoted the cytoskeleton recovery of steroid-induced TM cell damage, as confirmed by the pronounced upregulation of F-actin (**Figure 3F**). We concluded that the IOP-lowering effect of MET in this steroid OHT model can be largely explained by the attenuation of fibrotic alterations and rearrangement of the cell skeleton at sites of TM or trabecular outflow pathways.

### Protective effects of MET on TM in vitro

To test the effect of MET on TM in vitro, we performed studies on human TM cells (HTMCs). Western blotting (WB) showed that the expression of myocilin, a glucocorticoid-inducible gene in HTMCs, increased after DEX treatment (**Figure 4B**), confirming that the cells had characteristics of TM cells.

Higher doses (≥10 mM) of MET decreased the number of HTMCs, but this inhibitory effect was less evident at lower doses (<10 mM) (**Figure 4C**). To verify the protective role of MET in HTMCs, cells

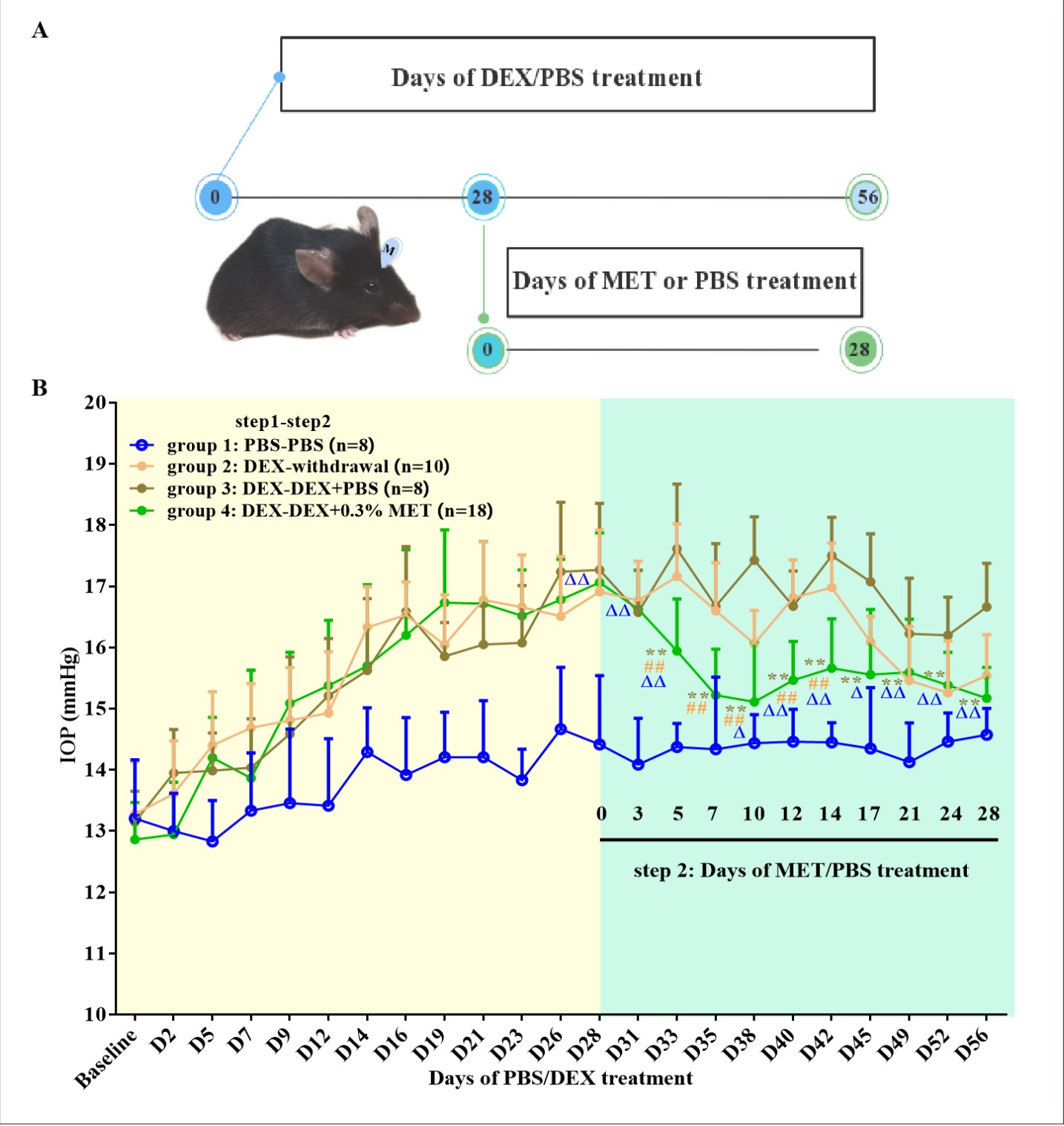

**Figure 2.** Effect of MET on OHT mouse models. (**A**) Experimental process overview. (**B**) MET effectively reversed the IOP in steroid-induced OHT mouse models. **p < 0.01 (comparison between groups 3 and 4), ##p < 0.01 (comparison between groups 2 and 4), △p < 0.05, and △△p < 0.01 (comparison between groups 1 and 4). DEX: dexamethasone, MET: metformin, PBS: phosphate-buffered saline, IOP: intraocular pressure, OHT: ocular hypertension.

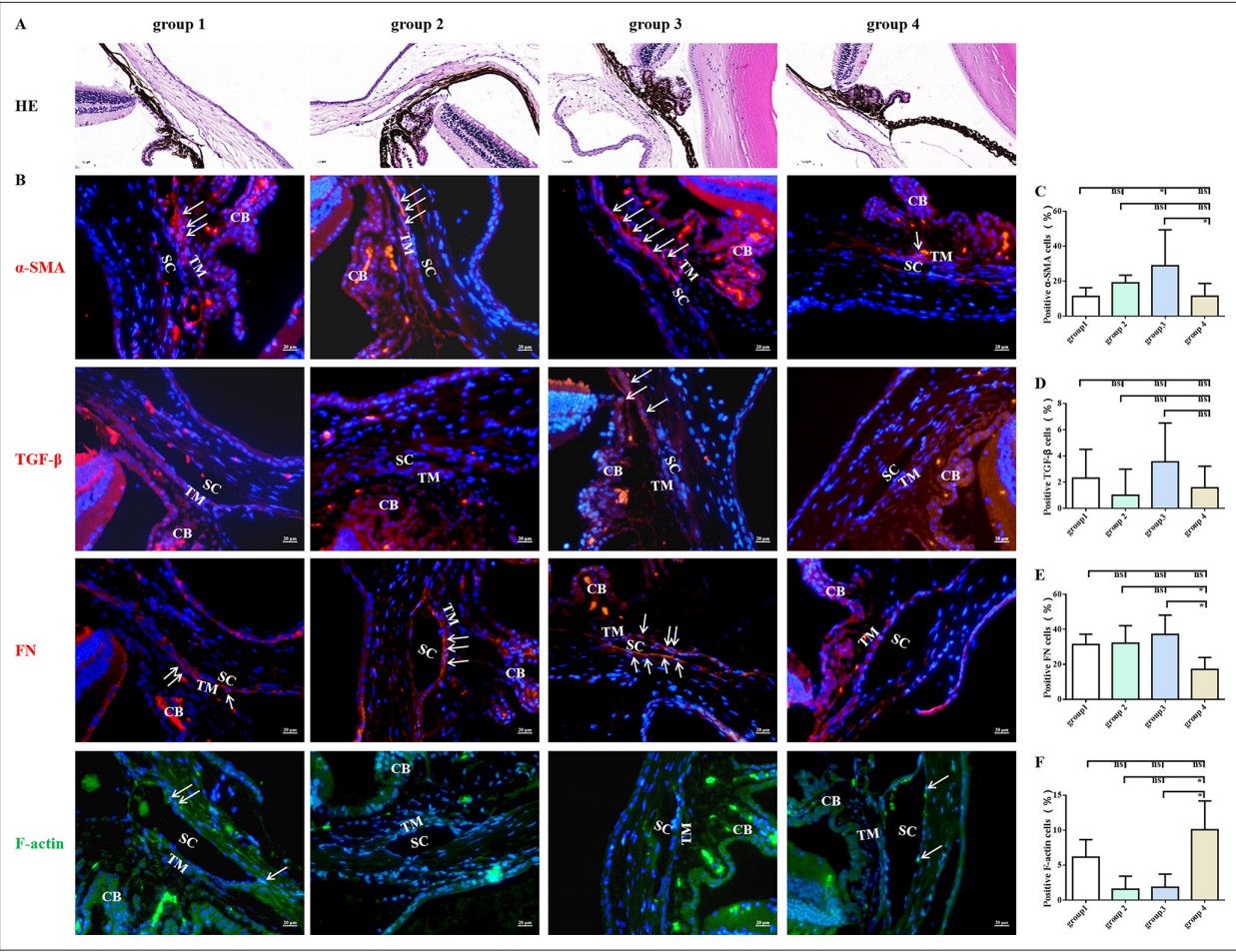

**Figure 3.** MET decreased the expression of fibrotic markers in steroid-induced trabecular meshwork stiffening in mice. Representative images of HE (**A**) and fibrotic markers (**B**). Scale bar, 20 μm, (**C**) Quantification of α-SMA of the models. (**D**) Quantification of TGF-β of the models after 4 weeks of DEX withdrawal or MET/PBS treatment. (**E**) Quantification of fibronectin (FN) of the models after 4 weeks of DEX withdrawal or MET/PBS treatment. (**F**) Quantification of F-actin of the models after 4 weeks of DEX withdrawal or MET/PBS treatment. *p < 0.05, ns: non-significance, DEX: dexamethasone, HE: haematoxylin and eosin, MET: metformin, PBS: phosphate-buffered saline, α-SMA: α-smooth muscle actin, TGF-β: transforming growth factor-β. White arrows indicate the representative positive cells.

were pre-treated with 100 μM tBHP for 1 hr to induce chronic oxidative stress (*Tang et al., 2013*), and subsequently exposed to low doses of MET (3, 5, and 10 mM) (shortened as L-MET in this study). The results showed that L-MET significantly reversed the inhibitory effect induced by tBHP (*Figure 4A, D*).

Furthermore, cells treated with L-MET showed less intracellular ROS signals compared to the control, and 1 mM MET reduced tBHP-induced ROS production. These results were confirmed by flow cytometry (FCM) analysis (*Figure 4E,F*) and further evidenced in DEX-induced (500 nM, 7 days) ROS increase in HTMCs (*Figure 4—figure supplement 1*). These results indicated that MET reversed oxidative damage to HTMCs.

## MET restored tBHP-induced cytoskeletal destruction in HTMCs and activated integrin/ROCK signals

Actin filaments play key roles in cortical polarisation and asymmetric spindle localisation during phagocytosis in HTMCs. F-actin was stained to determine whether actin dynamics were involved in tBHP-induced cellular dysfunction. As shown in *Figure 5A*, F-actin was evenly accumulated in the cytoplasm with a robust fluorescent signal in control HTMCs. Exposure to L-MET did not change the morphology of HTMCs as observed by inverted phase-contrast microscopy. Contrarily, HTMCs exposed to 1 hr of tBHP displayed intermittent distribution of actin filaments with faded fluorescent signals. Additionally, tBHP treatment significantly destroyed HTMCs, manifesting as various morphology-aberrant

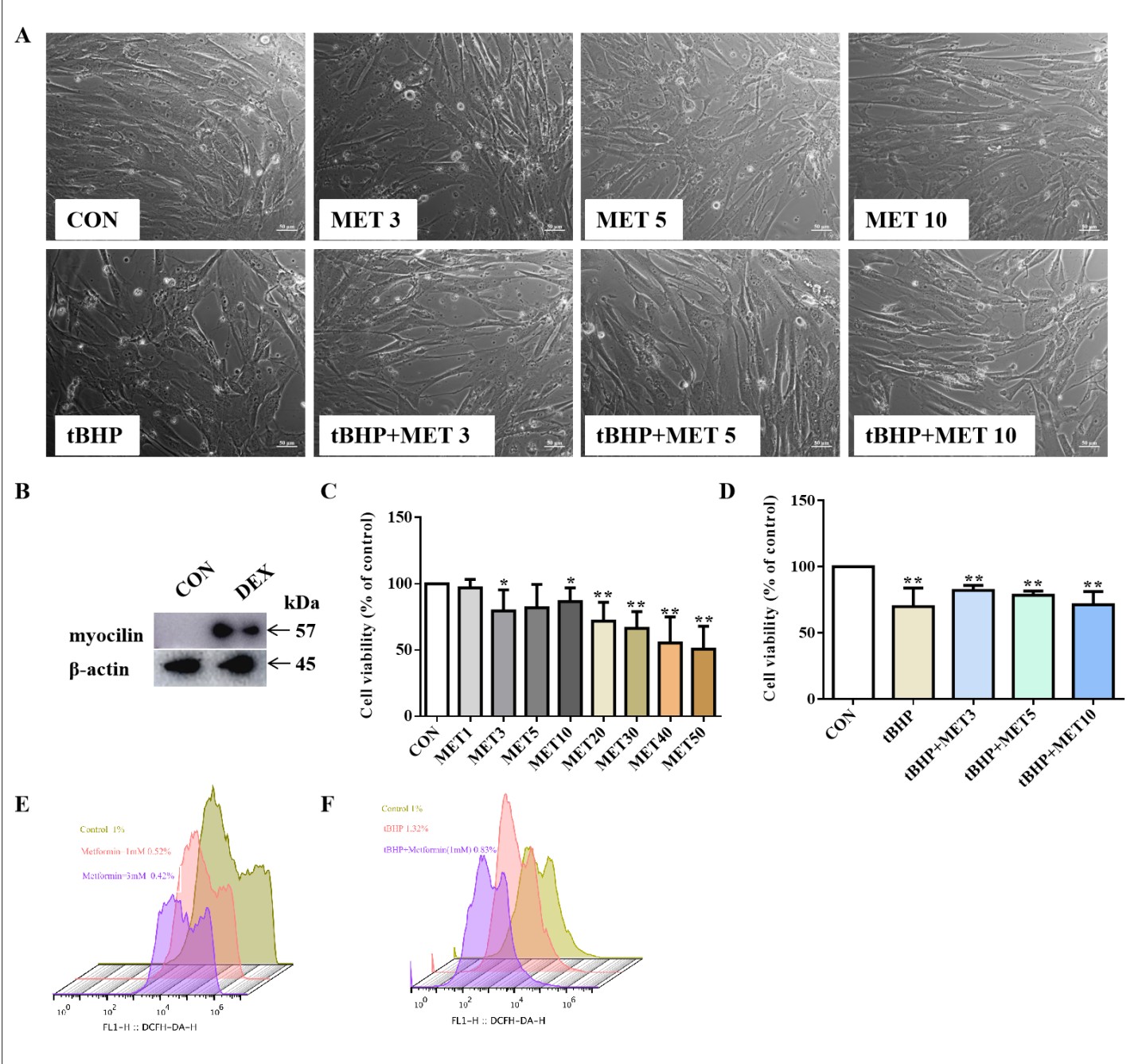

**Figure 4.** Low dose of MET reversed the disarranged morphology of HTMCs. (**A**) HTMCs were treated with MET for 24 hr with or without pre-treatment of tBHP for 1 hr. Representative images of cell distribution and morphology photographed by inverted microscopy. Scale bar, 50 μm, (**B**) The expression of myocilin after DEX treatment in HTMCs. (**C, D**) The relative HTMC viability after exposure to MET with different concentrations. Cell proliferation was measured using the CCK8 assay. (**E, F**) The relative ROS levels were assayed via flow cytometry, and the results showed that MET reduced the ROS production of HTMC induced by tBHP. *p < 0.05, **p < 0.01. DEX: dexamethasone, HTMCs: human trabecular meshwork cells, MET: metformin, ROS: reactive oxygen species, tBHP: tert-butyl hydroperoxide, TM: trabecular meshwork.

The online version of this article includes the following figure supplement(s) for figure 4:

**Figure supplement 1.** L-MET attenuated the ROS production in HTMCs induced by DEX treatment.

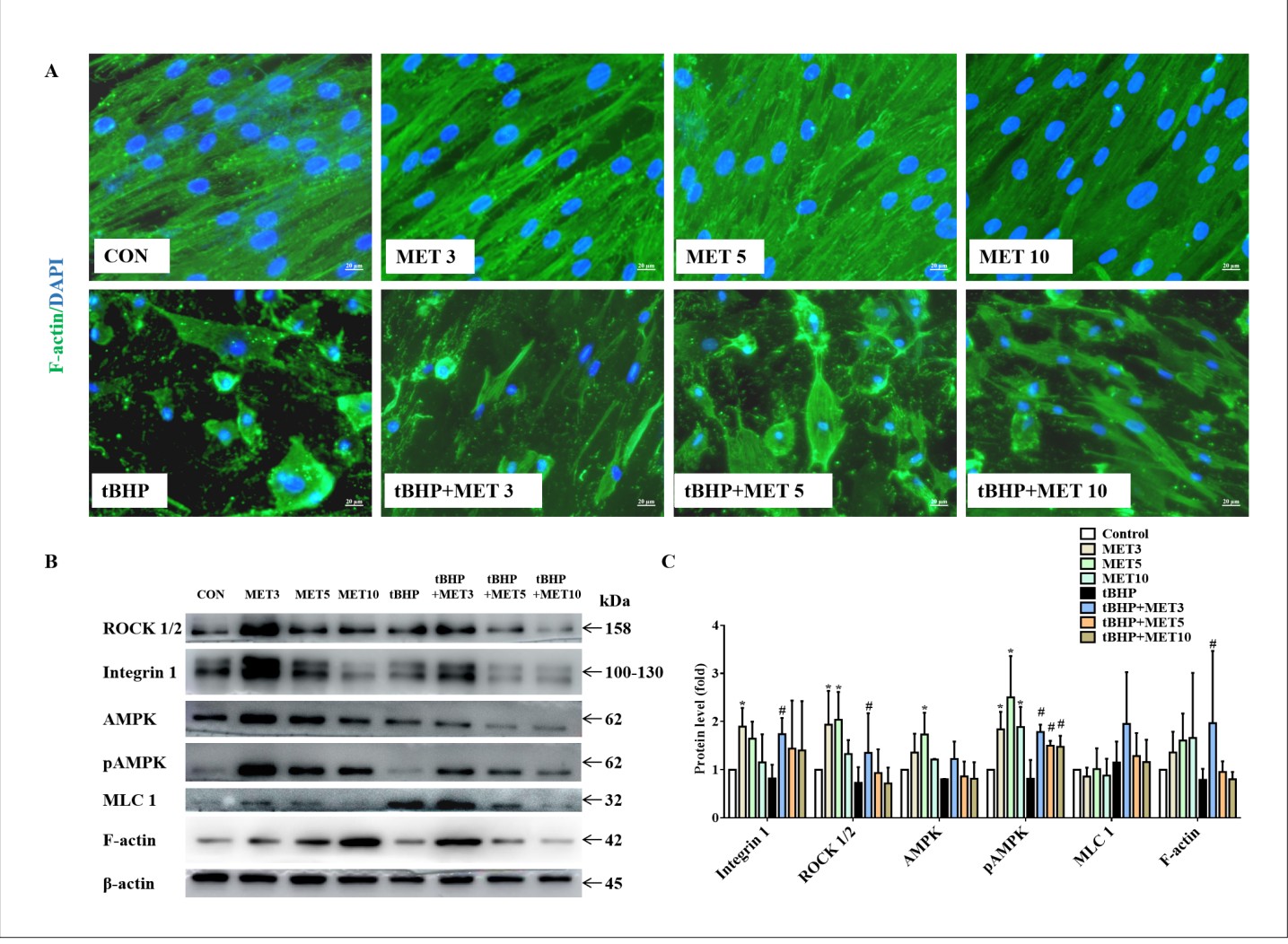

**Figure 5.** MET promoted the recovery of tBHP-induced cytoskeletal destruction (**A**) and activated the intergrin/ROCK pathway (**B**) in HTMCs. Scale bar, 20 μm. (**C**) The quantitative protein levels of B. *p < 0.05 (comparison with the control), #p < 0.05 (comparison with the tBHP-treated group). HTMCs: human trabecular meshwork cells, MET: metformin, tBHP: tert-butyl hydroperoxide.

cells with misaligned cytoplasm. These pathological changes could be partially rescued by L-MET, implying that MET could restore the dynamic instability from oxidative damage. Furthermore, WB results (**Figure 5B, C**) showed that L-MET significantly activated the integrin/ROCK pathway by upregulating integrin, ROCK, AMPK, and pAMPK in healthy HTMCs, and significantly activated integrin, ROCK, pAMPK, and F-actin in damaged HTMCs, and it was more pronounced in the 3 mM doses.

The ultrastructure of HTMCs was examined by transmission electron microscopy (TEM) (**Figure 6**). We observed a significant reduction in the amount and density of microfilaments in the tBHP-treated cells, and these changes recovered after L-MET treatment.

## Discussion

The major observations of the current study were the three significant phenotypic changes induced by MET in OHT mouse eyes and HTMCs. The first was the IOP-lowering effect in steroid-induced OHT mouse eyes. The second was reversal of the skeletal destruction of TM cells, both in vivo and in vitro. Third, there was a significant decrease in the accumulation of fibrotic markers, namely α-SMA and FN, in TM tissues. These results suggest a protective effect of MET in TM, probably via promoting cytoskeleton recovery through the integrin/ROCK pathway.

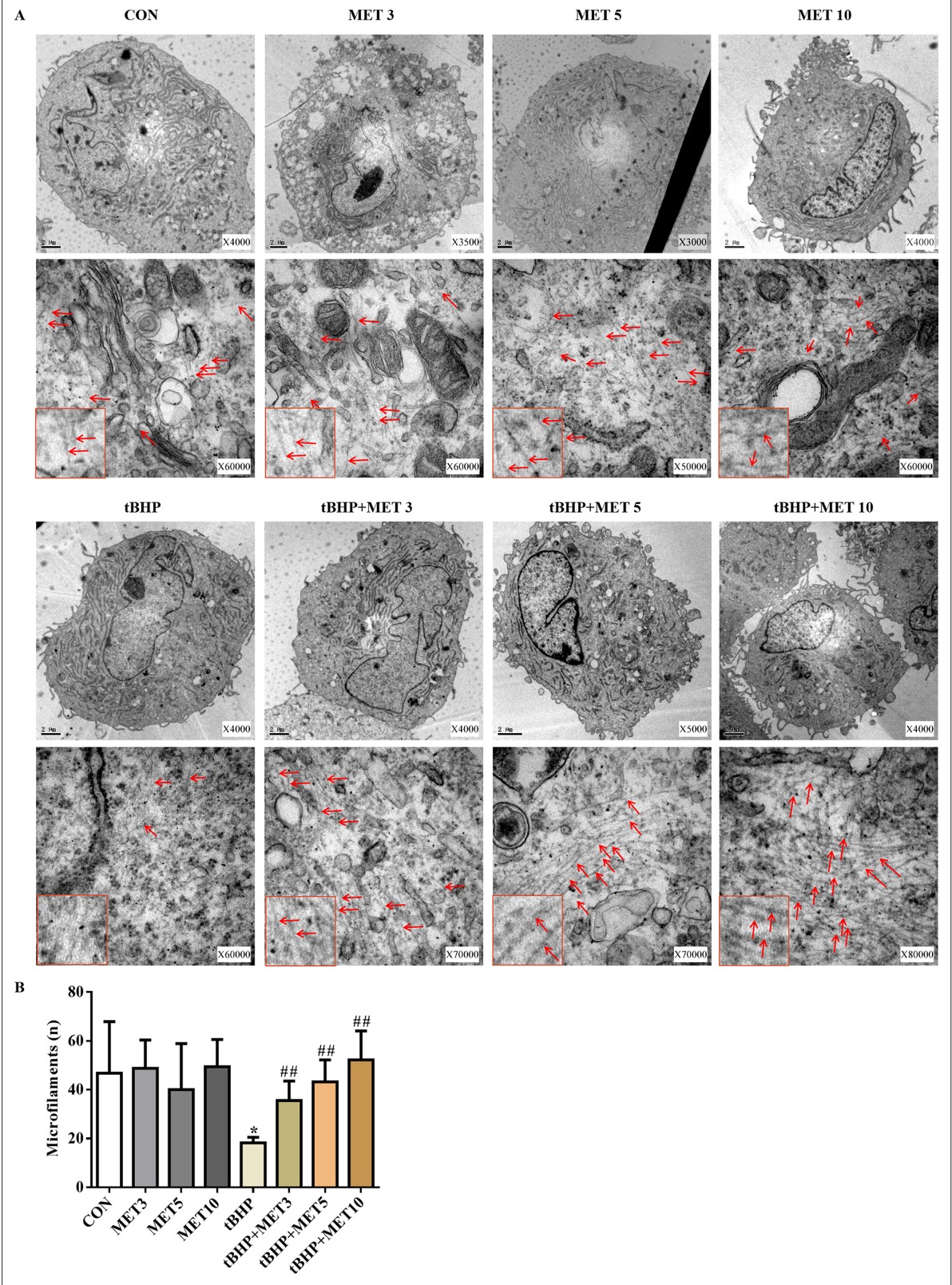

**Figure 6.** MET partially normalised the damaged microfilaments of HTMCs induced by tBHP (**A**). Scale bar, 2 μm, Red arrows indicate the representative microfilaments imaged by transmission electron microscopy. The red rectangles embedded in the lower left part of each picture are representative microfilaments with a magnification of ×150,000. (**B**) The quantitative microfilaments of A. *p < 0.05 (comparison with the control), ##p < 0.01 (comparison with the tBHP-treated group). HTMCs: human trabecular meshwork cells, MET: metformin, tBHP: tert-butyl hydroperoxide.

Steroid-induced OHT in mice was generated (*Wang et al., 2018*) with the characteristics of TM stiffening and elevated IOP (*Li et al., 2019*). In line with previous studies (*Clark et al., 2001*; *Johnson et al., 1997*; *Johnson et al., 1990*), our model suggested that the stiffness of AHO closely matched that of OAG in humans, including the deposition of extracellular matrix (ECM) in the TM, disordered cytoskeleton, increased AHO resistance, and elevated IOP. Similar to the study by *Zode et al., 2014*; *Ren et al., 2022*, our topical DEX elevated the IOP by 2.00–7.30 mm Hg after 4 weeks of induction.

TM tissues play an important role in the regulation of AHO function and IOP (*Wang et al., 2018*). TM cells have the properties of extracellular debris phagocytosis and ECM degradation. If these functions are disturbed, the balance of TM in keeping AHO free of obstructive debris is disrupted (*Last et al., 2011*). It is noteworthy that MET appeared to restore biomechanical properties of TM in our steroid-induced OHT model after 5 days of treatment, and this restoration was accompanied by significant downregulation of ECM proteins in TM.

Moreover, as evidenced by immunofluorescence (IF) and WB analyses, L-MET rearranged the disordered cytoskeleton of HTMCs. This finding was supported by a study conducted by *Li et al., 2020b*, who showed that 50 µM MET protected oocytes against cytoskeletal destruction. Interestingly, the time course of structural changes in the TM was consistent with the observed pharmacodynamics of MET on IOP, significantly decreasing 4 weeks of OHT because of MET treatment, which was consistent with the known effects of other ROCK regulators (*Lin et al., 2018*; *McMurtry et al., 2010*). Taken together, our results indicate that L-MET promotes the regeneration of abnormal HTMCs and enables the recovery of TM function; this should be confirmed in future studies.

As confirmed by WB analysis, complementation with L-MET treatment unregulated integrin, pAMPK, ROCK, and F-actin signals. These in vitro findings indicate that MET might elicit its protective effects through the regulation of the integrin/ROCK pathway in HTMCs. The in vivo observations in the OHT model again support the importance of the integrin/ROCK pathway in TM tissue protection. Generally, these signals mediate cellular biomechanical tension through actomyosin cytoskeletal tension, ECM synthesis, assembly, and degradation (*Nakajima et al., 2005*; *Pattabiraman and Rao, 2010*; *Rao et al., 2001*).

Inconsistent with the previous studies indicating that the AMPK activator suppressed ROCK activity, our study revealed a novel mechanism of MET action within which activation of both AMPK and ROCK signals sequentially upregulated F-actin expression in TM. We hypothesised that ROCK regulators protect TM via different mechanisms, either by inhibiting ROCK signals when increased AHO resistance is attributed to TM stiffness or by activating the ROCK pathway to promote cytoskeletal formation when it is damaged. In the current study, we presumed that MET lowered IOP through short-term changes in TM cell morphology and adhesion by regulating integrin/ROCK signalling. In addition, the steroid-induced OHT recovered its normal IOP with MET treatment, preventing further TM cell loss, whereas MET promoted the maintenance of cytoskeletal morphology and repair after injury; therefore, IOP was maintained at a normal level for an extended period.

Generally, these data emphasise the protective properties of MET in the conventional outflow pathway, consistent with a large body of literature on other tissues (*Rangarajan et al., 2018*; *Yi et al., 2021*; *Zhao et al., 2021*). In addition to MET-mediated changes in ECM turnover, the observed changes in ECM composition and amount may also be due to the MET-mediated opening of flow pathways and the consequential removal of ECM. Additionally, the observed reduction in ROS content in HTMC treated with L-MET was in line with earlier studies displaying its protective effect in several types of cells (*Ghasemnejad-Berenji et al., 2018*; *Huang et al., 2015*; *Louden et al., 2014*). In any case, therapy for steroid-induced OHT, which inhibits the cycle of damage and promotes cell remodelling by restoring function to a diseased tissue, offers a potential benefit for patients.

While experimental data were encouraging, there were limitations in our study. First, the concentration of MET in the anterior chamber was not measured. Additionally, we did not investigate its toxicity on ocular surfaces, such as tear film stability and components, which may change with the application of MET. Second, the sample size of the animals was relatively small, and a larger sample is needed to verify the protective effect of MET on TM. Third, the experimental period was too short to prove the long-term effects of MET and the presented figures were not in the same direction. Thus, additional studies are needed in the future.

In summary, promotion of TM cytoskeleton recovery by MET treatment may be an underlying mechanism for IOP reduction and AHO increase. A thorough investigation of the mechanisms by

which cytoskeleton recovery is promoted will improve our understanding of the therapeutic mechanism of MET. If MET is confirmed to promote damaged cytoskeleton recovery and maintenance in the human body, it may have clear therapeutic effects on the main sites of glaucoma pathogenesis, which will broaden their application in the field of glaucoma. In conclusion, we revealed that integrin/ROCK activation reprograms metabolism in TM cells by enhancing the cytoskeleton and downregulating excessive ECM proteins.

## Materials and methods

### Animals

All the animals were treated in accordance with the principles of the Declaration of Helsinki and in compliance with the Association for Research in Vision and Ophthalmology (ARVO) Statement. All experiments were approved by the Institutional Animal Care and Use Committee of the Wenzhou Medical University (wydw2022-0209). Healthy C57BL/6J mice (age: 6–8 weeks, male, weight: 20–25 g) were used in this study. The mice were purchased from Zhejiang Vital River Experimental Animal Technology Co. Ltd (Charles River Lab China), bred/housed in clear cages, and kept in housing rooms at 21°C with a 12:12 hr light:dark cycle.

### OHT animal model and drug treatments

The animal experiments were performed in two consecutive steps. In the first step, OHT C57BL/6J mouse model was induced by topical 0.1% DEX phosphate (dissolved in PBS twice daily, 8–9 AM and 5–6 PM) as described previously (*Li et al., 2021*), and sterile PBS was used as a vehicle control. The second step was conducted 4 weeks after steroid induction. The successfully created OHT mouse models with elevated IOP were randomly divided into three groups: DEX withdrawal (group 2, DEX was stopped without any additional prescribed drug, $n = 10$), DEX + PBS (group 3, $n = 8$), and DEX + 0.3% MET (group 4, $n = 18$). Mice in groups 3 and 4 received an additional 4 weeks of supplemented drug (PBS or 0.3% MET) delivered bilaterally with the continued synchronous use of steroid, leading to an entire 8-week duration of DEX treatment. Meanwhile, PBS-only treatment was administered for 8 weeks in the control (group 1, $n = 8$). Drugs were prescribed twice daily with at least 15-min interval when administering 2 types of eye drops to ensure effective penetration.

### IOP measurements

Briefly, mice were anaesthetised using gaseous isoflurane (approximately 2 min) and topical alcaine (Alcon-couvreur n. v, Rijksweg, Puurs Belgium). IOP was measured using a rebound tonometer (icare TONOVET; Vantaa, Finland). Each recorded IOP was the average of five measurements and three IOP readings were recorded for the same eye to calculate the mean value. IOP measurements were conducted every 2–3 days (Monday, Wednesday, and Friday, between 2 PM and 3 PM).

### Weight measurement

The anaesthetised mice were gently placed on digital electronic scales (Electronic Scale, Kunshan, China) to measure their weight. The effective reading of BW was recorded to an accuracy of 0.01 g. The BW of each mouse was calculated from the average of three test values.

### Cell culture and treatment

HTMCs were purchased from BNCC (338506, Shanghai, China), which were authenticated by STR profiling. We confirmed that no mycoplasma contamination was detected by regular examinations. Cells were cultured in Dulbecco's Modified Eagle's Medium: F-12 (DMEM/F12) (Cytiva, HyClone Laboratories, Logan, UT, USA) containing 10% foetal bovine serum (BI, USA) and antibiotics (100 U/ml penicillin and 100 μg/ml streptomycin, Gibco, Life Technologies Corporation, NY, USA) at 37°C and 5% $CO_2$. To identify the characteristics of HTMCs, cells were treated with 500 nM DEX (Shanghai Macklin Biochemical Co., Ltd, China) for 7 days, and the expression of myocilin was evaluated by WB (*He et al., 2019*). For drug testing, tBHP (Damas-beta, China) and MET (Sigma-Aldrich, St. Louis, USA) were dissolved in DMEM/F12. HTMCs were pre-treated with tBHP solution for 1 hr to induce oxidative damage, followed by 24-hr incubation in normal culture medium containing MET at certain concentrations.

## Cell viability

Cell viability was measured using the CCK8 assay kit (APE BIO), following the manufacturer's instructions. The cells were seeded in 96-well plates and exposed to increasing concentrations of MET (0, 1, 3, 5, 10, 20, 30, 40, and 50 mM) for 24 hr. The cell viability was determined by measuring the optical density at 490 nm using an absorbance microplate reader (SpectraMax 190; version 7.1.0, Molecular Devices, CA, USA).

## Detection of intracellular ROS levels

Intracellular ROS levels were determined using a 2′,7′-dichlorofluorescein diacetate (DCFH-DA) probe (Beyotime Biotechnology, Shanghai, China) by measuring the oxidative conversion of cell-permeable DCFH-DA to fluorescent dichlorofluorescein according to the manufacturer's instructions and a previous study (*Xu et al., 2020*). Briefly, 10 µM DCFH-DA solution diluted in serum-free DMEM/F12 medium was added and incubated with the cells, then they were kept in the dark for 20 min. Excess DCFH-DA was removed by washing the cells three times with serum-free cell culture medium. ROS was detected by both IF imaging and FCM (BD Accuri C6 Plus, CA, USA), and analysed by Flow Jo (Flowjo X 10.0.7r2). The values were normalised to signals from the control group.

## Western blotting

Cells were lysed 24 hr after drug treatment as previously described (*Xu et al., 2020*). After gel separation and membrane transfer, myocilin (Abcam), integrin 1 (Abcam), AMPK (CST), pAMPK (CST), ROCK1/2 (Abcam), MLC1 (Abcam), and F-actin (CST) were detected. β-Actin (CST) was used as the loading control. Western blot membranes were developed using a chemiluminescence detection system (Amersham Imager 680RGE; GE Healthcare Bio-Sciences AB, Sweden, Japan).

## Histology, immunostaining, and TEM

At the time of harvest, the mice were re-anaesthetised; the eyes were fixed in 4% paraformaldehyde overnight and embedded in paraffin or optimum-cutting temperature compound in the sagittal axis. The sections were incubated overnight with primary antibodies at 4°C according to the manufacturer's instructions (*Xu et al., 2020*). The secondary antibodies were goat anti-rabbit or anti-mouse (CST) at a 1:500 dilution. Sections were subsequently incubated with 2-(4-amidinophenyl)-6-indolecarbamidine dihydrochloride (DAPI, Beyotime Institute of Biotechnology, Shanghai, China) for 5 min to stain the nuclei, washed, and then mounted.

The cells were fixed with 4% paraformaldehyde for 15 min, permeabilised with 0.25% Triton X-100 for 20 min, and blocked with bovine serum albumin for 30 min at room temperature. After blocking, they were incubated overnight with primary antibodies at 4°C. The next day, the cells were rinsed and incubated with secondary antibodies conjugated to Alexa Fluor (CST) for 1 hr at room temperature. Fluorescent images were obtained using an inverted microscopy (Carl Zeiss Microscopy GmbH, Göttingen, Germany) with the luminescence of X-Cite Series 120(Lumen Dynamics Group Inc, Canada).

For electron microscopy studies, cells were fixed in 2.5% glutaraldehyde (Shanghai Macklin Biochemical Co., Ltd, China) and embedded in Epon resin, and 80-nm sagittal thin sections were cut through iridocorneal tissues using an ultramicrotome (Power Tome-XL, RMC Products, USA). Sections were stained with uranylacetate/lead citrate and examined with a transmission electron microscope (HITACHI, H-7500).

## Image analysis

Immunohistochemical images were obtained using a microscope (ECLIPSE 80i, Nikon) and analysed using the NIS-Elements Imaging Software (3.22.00; Build 700, LO, USA). For quantification, high-power fields (×400 magnification) of the AHO from each model were captured. TM region was defined as the area between the ciliary body and the inner wall of Schlemm's canal, with anterior and posterior end points of Schwalbe line and scleral spur, respectively (*Yan et al., 2022*). For microfilament quantification, five fields with ×80,000 magnification of each group were captured with TEM, within which two were from the cell cortex region (defined as within 400 nm beneath the cell membrane) (*Svitkina, 2020*) and the remaining three were beyond this area. ImageJ (v1.52a, National Institutes of Health, USA) was used to quantify the positively stained cells, which referred to those cells stained by both blue colour from DAPI and green or red signals from the secondary antibodies.

## Statistical analysis

Data analysis was conducted using SPSS (version 22.0) software and GraphPad Prism (version 7.0). Cell- and animal-based experiments included at least three biological and technical replicates. Continuous data are summarised as mean ± standard deviation. Student's $t$-test and analysis of variance were used to test the differences in continuous variables. A paired $t$-test was used to compare IOP changes from the baseline. Statistical significance was set at p value <0.05.

## Acknowledgements

We thank Mao Huiyan, Li Jinxin, and Zhou Mengtian, who helped with the delivery of eye drops in the mice. We also acknowledge Fang Zhouxi and Pan Liangliang for their assistance with TEM. This work was supported by the National Natural Science Foundation of China (Grant No. 82201177), Health Commission of Zhejiang Province (Grant No. 2023KY914), Key R&D Program of Zhejiang (2022C03112), Leading Scientific and Technological Innovation Talents in Zhejiang Province (2021R52012), National Key R&D Program of China (2020YFC2008200), Zhejiang Provincial National Science Foundation of China (LQ18H120010), Key Innovation and Guidance Program of the Eye Hospital, School of Ophthalmology & Optometry, Wenzhou Medical University (YNZD 2201903), and the Wenzhou Municipal Technological Innovation Program of High-level Talents (No. 604090352/577).

---

## Additional information

### Funding

| Funder | Grant reference number | Author |
|---|---|---|
| National Natural Science Foundation of China | 82201177 | Lijuan Xu |
| Health Commission of Zhejiang Province | 2023KY914 | Lijuan Xu |
| Key R&D Program of Zhejiang | 2022C03112 | Yuanbo Liang |
| Leading Scientific and Technological Innovation Talents in Zhejiang Province | 2021R52012 | Yuanbo Liang |
| National Key Research and Development Program of China | 2020YFC2008200 | Yuanbo Liang |
| Zhejiang Provincial National Science Foundation of China | LQ18H120010 | Yuanbo Liang |
| Key Innovation and Guidance Program of the Eye Hospital, School of Ophthalmology & Optometry, Wenzhou Medical University | YNZD 2201903 | Yuanbo Liang |
| Wenzhou Municipal Technological Innovation Program of High-level Talents | 604090352/577 | Yuanbo Liang |

The funders had no role in study design, data collection, and interpretation, or the decision to submit the work for publication.

### Author contributions

Lijuan Xu, Yuanbo Liang, Conceptualization, Resources, Data curation, Software, Formal analysis, Supervision, Funding acquisition, Validation, Investigation, Visualization, Methodology, Writing

- original draft, Project administration, Writing - review and editing; Xinyao Zhang, Xiaorui Gang, Data curation, Formal analysis, Investigation, Methodology; Yin Zhao, Formal analysis, Investigation, Methodology; Tao Zhou, Data curation, Investigation, Methodology; Jialing Han, Yang Cao, Binyan Qi, Shuning Song, Investigation, Methodology; Xiaojie Wang, Investigation

### Author ORCIDs
Lijuan Xu (iD) http://orcid.org/0000-0003-0361-2611
Yuanbo Liang (iD) http://orcid.org/0000-0001-9685-7356

### Ethics
The Institutional Animal Care and Use Committee of the Wenzhou Medical University (wydw2022-0209).

### Decision letter and Author response
Decision letter https://doi.org/10.7554/eLife.81198.sa1
Author response https://doi.org/10.7554/eLife.81198.sa2

## Additional files

### Supplementary files
• Source data 1. Source data files for *Figure 1*, *Figure 2*, *Figure 3*, *Figure 4*, *Figure 5*, *Figure 6* and *Figure 4—figure supplement 1*.
• MDAR checklist
• Reporting standard 1. The ARRIVE guidelines 2.0: author checklist.

### Data availability
All data generated or analysed during this study are included in the manuscript and supporting files. Source data files have been provided for Figures 1–6 and Figure 4—figure supplement 1.

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
