## [Editor Report]

The claims that metformin reduced elevated intraocular pressure in vivo and regulated cytoskeleton remodeling in vitro are supported by convincing data. This study provides a new direction for further exploration toward the treatment of primary open-angle glaucoma.

---

## [Decision Letter]

**Decision letter after peer review:**

Thank you for submitting your article "Metformin protects trabecular meshwork against oxidative injury via activating integrin/ROCK signals" for consideration by *eLife*. Your article has been reviewed by 2 peer reviewers, including Mishaela R Rubin as the Reviewing Editor and Reviewer #1, and the evaluation has been overseen by a Reviewing Editor and Mone Zaidi as the Senior Editor.

Essential revisions:

1) Please provide control data

2) Please provide information about animal numbers, sexes, biological repeats, and technical repeats

3) Please provide information about the duration of dex treatment

4) Please consider providing data using dex-treated cells to detect a metformin anti-oxidant effect

*Reviewer #1 (Recommendations for the authors):*

In the Introduction, consider removing sentences about the cost and availability of other ROCKi, as these are specific to a geographical location and distracting from the scientific rationale.

In Figure 2B, it looks like IOP might be rising in the MET groups towards the last few days. If possible, please provide data on the mice beyond D42.

There should be mouse control that was not steroid-treated. This would allow comparison of the parameters in the metformin-treated mice with controls.

The authors state (line 160) that the inhibitory effect of MET was not evident at lower doses, but in Figure 4C it looks like 3 mM decreased the number of HTMC.

What were the ROS signals that were measured by FCM (Figure 4E-F)?

In Figure 5, ROCK did not increase statistically in tBHP+MET at any dose, but the text states otherwise (line 192).

In Discussion, line 270, the wording is confusing- "recovery to normal IOP prevents further TM cell loss, whereas metformin…." Does this mean that IOP mice recovered in the absence of metformin?

*Reviewer #2 (Recommendations for the authors):*

Although the presented results were promising with translational potential in the future, there are some concerns as listed below:

1. The animal numbers and sexes in each group and the biological repeats and technical repeats were not clearly listed which must be listed in the manuscript as well as in the figure legends.

2. Page 2 Abstract: Methods: Subsequent TEM tests were not mentioned.

3. Page 4 lines 64-65: "Theoretically, TM-targeting drugs are potentially effective in lowering IOP caused by diseased TM." This is not clear what the authors meant. Please modify and add references.

4. Page 5 Introduction: Please briefly introduce the effect of MET on ROS.

5. For all the sections with immunofluorescent staining, it's not clear how the authors defined the TM region. They didn't show any specific marker expression and didn't have the same direction as all the figures.

6. Figure 1. How did the authors calculate the a-SMA and TGFb positive cells, especially TGFb is an ECM marker

7. How long did Dex treatment last? Was it stopped at 3 weeks or was it continued together with metformin treatment? If continued, how did they give the different treatments? Please describe those in detail. If Dex was not continued, IOP could possibly reduce in 3 weeks. Please compare if IOP in PBS group has statistically changed or not with that when Dex stropped.

8. Figure 2, several other control groups should also be compared, like PBS only, PBS+metformin eyedrops, and PBS+metformin oral without Dex treatment.

9. Besides the in vitro tBHP induced oxidative cell model, it would be more convincing if the authors could also use Dex-treated cells to detect the metformin anti-oxidation effect since they used Dex induced mouse model.

10. Figure 4, how many biological repeats and technical repeats? One of the characteristics of TM cells is that TM cells have increased expression of myocilin after exposure to Dex. Figure 4B didn't show myocilin expression increasing after Dex.

11. Fig6, it's not clear what the authors marked. How did they mark the microfilaments? A statistical analysis with graphs would be more clear and convincing.

12. The discussion of this study lacks further elaboration. There was no mention of the limitation of the study.

---

## [Author Response]

Essential revisions:1) Please provide control data.

Thank you for this important suggestion. We have conducted a repeated experiment during the submission period and have extended both the OHT modelling process and drugs’ IOP-lowering effect tests from 21 to 28 days to strengthen the power of this study. Metformin oral group was deleted because of the unstable intake amount, which may reduce the reliability of the results. New data, including the control data, have been added to the manuscript. The main points are summarised as follows:

Materials and methods (p. 10, Line 8-22 in the manuscript)

“OHT animal model and drug treatments

The animal experiments were performed in two consecutive steps. In the first step, OHT C57BL/6J mouse model was induced by topical 0.1% DEX phosphate (dissolved in PBS twice daily, 8–9 AM and 5–6 PM) as described previously (*Li, et al.*,*2021*), and sterile PBS was used as a vehicle control. The second step was conducted 4 weeks after steroid induction. The successfully created OHT mouse models with elevated IOP were randomly divided into three groups: DEX withdrawal (group 2, DEX was stopped without any additional prescribed drug, n=10), DEX+PBS (group 3, n=8), and DEX+0.3% MET (group 4, n=18). Mice in groups 3 and 4 received an additional 4 weeks of supplemented drug (PBS or 0.3% MET) delivered bilaterally with the continued synchronous use of steroid, leading to an entire 8-week duration of DEX treatment. Meanwhile, PBS-only treatment was administered for 8 weeks in the control (group 1, n=8). Drugs were prescribed twice daily with at least 15-min interval when administering 2 types of eye drops to ensure effective penetration.”

Results (p. 5, Line 4-27 in the manuscript, Figure 2 and 3 in the figure legends)

“MET effectively reversed steroid-induced OHT in mice

In this step, successfully DEX-induced OHT mouse models were randomly assigned into three groups with the continued prescription of DEX: DEX withdrawal (group 2, n=10), DEX+PBS (group 3, n=8), and DEX+0.3% MET (group 4, n=18). Meanwhile, a group of mice were treated with solo PBS as control with a duration of 8 weeks (group 1, n=8).

The experimental procedure is illustrated in Figure 2A. The IOPs after 28 days of steroid induction in all three OHT groups were similar (17.27±1.08, 16.91±1.01, and 17.06±0.82 mmHg, respectively). Compared with group 3, DEX-withdrawal mice had significantly reduced IOP after 17–28 days (all *p* < 0.05). After 5 days of MET treatment, IOPs significantly reduced in group 4 (*p* < 0.01, Figure 2B). In fact, MET almost completely neutralised steroid-induced OHT, returning IOP to near DEX-withdrawal levels on day 21, suggesting a therapeutic role of MET in OAG. Specific IOP values are shown in Figure 2B.”

**“**Figure 2. Effect of MET on OHT mouse models. A. Experimental process overview. B. MET effectively reversed the IOP in steroid-induced OHT mouse models. ***p* < 0.01 (comparison between groups 3 and 4), ## *p* < 0.01 (comparison between groups 2 and 4), △*p* < 0.05 and △△*p* < 0.01 (comparison between groups 1 and 4). DEX: dexamethasone, MET: metformin, PBS: phosphate buffered saline, IOP: intraocular pressure, OHT: ocular hypertension”

“MET attenuated the steroid-induced TM cytoskeletal destruction in vivo

As shown in Figure 3, MET improved fibrosis and the intensity of phalloidin labelling of F-actin in the TM tissue of steroid-induced OHT C57BL/6 mice. Quantitative comparison showed a significant difference in the number of α-SMA- and fibronectin (FN)-positive TM cells between MET-treated and control mice (Figure 3C, 3E). Additionally, MET treatment promoted the cytoskeleton recovery of steroid-induced TM cell damage, as confirmed by the pronounced upregulation of F-actin (Figure 3F). We concluded that the IOP-lowering effect of MET in this steroid OHT model can be largely explained by the attenuation of fibrotic alterations and rearrangement of the cell skeleton at sites of TM or trabecular outflow pathways.”

Figure 3. MET decreased the expression of fibrotic markers in steroid-induced trabecular meshwork stiffening in mice. A-B. Representative images of HE (A) and fibrotic markers (B). C. Quantification of α-SMA of the models. D. Quantification of TGF-β of the models after 4 weeks of DEX withdrawal or MET/PBS treatment. E. Quantification of fibronectin (FN) of the models after 4 weeks of DEX withdrawal or MET/PBS treatment. F. Quantification of F-actin of the models after 4 weeks of DEX withdrawal or MET/PBS treatment. **p* < 0.05, ns: non-significance, DEX: dexamethasone, HE: haematoxylin and eosin; MET: metformin, PBS: phosphate buffered saline. White arrows indicate the representative positive cells.”

2) Please provide information about animal numbers, sexes, biological repeats, and technical repeats.

Thanks for the suggestion. We have added the animal number in the manuscript (see p. 5, Line 6-9 or “Essential revisions 1”). The animal sexes were shown in p. 10, Line 4-5: “ Healthy C57BL/6J mice (age: 6–8 weeks, male, weight: 20–25 g) were used in this study”. Biological and technical repeats were shown in p. 13, Line 21-22: “Cell-based and animal-based experiments included at least three biological and technical replicates.”

3) Please provide information about the duration of dex treatment.

Thanks. We have added the information as follows (see p. 10, Line 16-19 in the manuscript or “Essential revisions 1”):

“Mice in groups 3 and 4 received an additional 4 weeks of supplemented drug (PBS or 0.3% MET) delivered bilaterally with the continued synchronous use of steroid, leading to an entire 8-week duration of DEX treatment.”

4) Please consider providing data using dex-treated cells to detect a metformin anti-oxidant effect.

Thanks for the suggestion. We have conducted the DEX-treated HTMC to evaluate the anti-oxidant effect of metformin. We found that after DEX treatment, the DCF fluorescence intensity was significantly increased in HTMCs, and this effect was attenuated by the additional treatment of low doses metformin. This part of result was presented as supplemental Figure 1 for Figure 4:

“Figure S1. L-MET attenuated the ROS production in HTMCs induced by DEX treatment. A. ROS signals of HTMCs were determined by labelling cells with DCFH-DA and photographed by an inverted phase-contrast microscopy. B. Quantification of DCF-positive cells in A. C. The relative ROS levels assayed by flow cytometry showed that 3mM MET reduced the ROS production of HTMC induced by DEX. ***p* < 0.01, **p* < 0.05, ns: non-significance. DEX: dexamethasone, MET: metformin, ROS: reactive oxygen species, HTMCs: human trabecular meshwork cells, L-MET: low doses of metformin (≤ 10 mM)”

Reviewer #1 (Recommendations for the authors):In the Introduction, consider removing sentences about the cost and availability of other ROCKi, as these are specific to a geographical location and distracting from the scientific rationale.

Thanks for the comments. We agree with the reviewer’s opinion and have deleted the above sentences from the text.

In Figure 2B, it looks like IOP might be rising in the MET groups towards the last few days. If possible, please provide data on the mice beyond D42.

Thank you for your advice. We have extended the experimental period of drugs’ IOP-lowering effect from 21 days to 28 days, and the new data have been provided in the manuscript (see Figure 2 legend, or “Essential revisions 1”).

There should be mouse control that was not steroid-treated. This would allow comparison of the parameters in the metformin-treated mice with controls.

Thank you for your advice. We have extended the experimental period of drugs’ IOP-lowering effect from 21 days to 28 days, and the new data have been provided in the manuscript (see Figure 2 legend, or “Essential revisions 1”).

The authors state (line 160) that the inhibitory effect of MET was not evident at lower doses, but in Figure 4C it looks like 3 mM decreased the number of HTMC.

Thanks. We have revised the manuscript accordingly (see p. 6, Line 3-4 in the manuscript):

“Higher doses (≥10 mM) of MET decreased the number of HTMCs, but this inhibitory effect was less evident at lower doses (<10 mM) (Figure 4C).”

What were the ROS signals that were measured by FCM (Figure 4E-F)?

Thanks. Intracellular ROS levels were determined by measuring the oxidative conversion of cell-permeable DCFH-DA to fluorescent DCF. The ROS signals were referred to positive DCF cells analysed by FCM, and the explanation was added in the manuscript (p. 11, Line 25-30, p. 12, Line 1-4):

“Intracellular ROS levels were determined using a 2′,7′-dichlorofluorescein diacetate (DCFH-DA) probe (Beyotime Biotechnology, Shanghai, China) by measuring the oxidative conversion of cell-permeable DCFH-DA to fluorescent dichlorofluorescein according to the manufacturer’s instructions and a previous study (*Xu, et al.*,*2020*). Briefly, 10 μM DCFH-DA solution diluted in serum-free DMEM/F12 medium was added and incubated with the cells, then they were kept in the dark for 20 min. Excess DCFH-DA was removed by washing the cells three times with serum-free cell culture medium. ROS was detected by both IF imaging and FCM (BD Accuri C6 Plus, CA, USA), and analysed by Flow Jo (Flowjo X 10.0.7r2).”

In Figure 5, ROCK did not increase statistically in tBHP+MET at any dose, but the text states otherwise (line 192).

Thanks for this important comment. We have replicated the experiment, analysed, and modified the text to match the Figure accordingly. Modified texts are shown as follows (p. 6, Line 28-30, p. 7, Line 1 in the manuscript):

“Furthermore, WB results (Figure 5B-C) showed that L-MET significantly activated the integrin/ROCK pathway by upregulating integrin, ROCK, AMPK, and pAMPK in healthy HTMCs, and significantly activated integrin, ROCK, pAMPK, and F-actin in damaged HTMCs, and it was more pronounced in the 3 mM doses.”

In Discussion, line 270, the wording is confusing- "recovery to normal IOP prevents further TM cell loss, whereas metformin…." Does this mean that IOP mice recovered in the absence of metformin?

Thanks. We have made revision accordingly for a clearer and more accurate expression (see p. 8, Line 26-29 in the manuscript):

“In addition, the steroid-induced OHT recovered its normal IOP with MET treatment, preventing further TM cell loss, whereas MET promoted the maintenance of cytoskeletal morphology and repair after injury; therefore, IOP was maintained at a normal level for an extended period.”

Reviewer #2 (Recommendations for the authors):Although the presented results were promising with translational potential in the future, there are some concerns as listed below:1. The animal numbers and sexes in each group and the biological repeats and technical repeats were not clearly listed which must be listed in the manuscript as well as in the figure legends.

Thanks. We have added the animal number in the manuscript (see p. 5, Line 6-9 or “Essential revisions 1”). The animal sexes were shown in p. 10, Line 4-5: “Healthy C57BL/6J mice (age: 6–8 weeks, male, weight: 20–25 g) were used in this study”. Biological and technical repeats were shown in p. 13, Line 21-22: “Cell-based and animal-based experiments included at least three biological and technical replicates.”

2. Page 2 Abstract: Methods: Subsequent TEM tests were not mentioned.

Thanks. We have added the information as follows (see p. 2, Line 13-14 in the manuscript):

“Transmission electron microscopy was performed to observe microfilaments in HTMCs.”

3. Page 4 lines 64-65: "Theoretically, TM-targeting drugs are potentially effective in lowering IOP caused by diseased TM." This is not clear what the authors meant. Please modify and add references.

Thanks for the suggestion. We have revised the text accordingly (see p. 3, Line 12-13 in the manuscript):

“Theoretically, drugs promoting the recovery of damaged TM are potentially effective in lowering IOP.”

4. Page 5 Introduction: Please briefly introduce the effect of MET on ROS.

Thanks for this suggestion. We have summarised the effect of MET on ROS as follows (see p. 4, Line 4-11 in the manuscript):

“Reduction of intracellular ROS levels by MET via activating AMPK signal has been reported in primary hepatocytes (*Ota, et al.*,*2009*), vestibular cells (*Lee, et al.*,*2014*), and human immune cells (CD14^+^ monocytes, CD3^+^ T cells, CD19^+^ B cells, and CD56^+^ NK cells) (*Hartwig, et al.*,*2021*). Conversely, there is also evidence on cellular ROS level increase in some cancer cells after MET treatment, including AsPC-1 pancreatic (*Warkad, et al.*,*2021*), osteosarcoma (*Li, et al.*,*2020*), and breast cancer cells (*Yang, et al.*,*2021*). These seemingly contradictory results suggested that MET plays different roles under different metabolic environments.”

5. For all the sections with immunofluorescent staining, it's not clear how the authors defined the TM region. They didn't show any specific marker expression and didn't have the same direction as all the figures.

Thanks. We have added the definition of the TM region in the Method section and marked TM (trabecular meshwork), SC (Schlemm’s canal), CB (ciliary body), and representative positively stained cells in all immunofluoresence figures (see p. 13, Line 10-12 in the manuscript).

“TM region was defined as the area between the ciliary body and the inner wall of Schlemm’s canal, with anterior and posterior end points of Schwalbe line and scleral spur, respectively (*Yan, et al.*,*2022*).”

The non-conformity with the figures’ direction was listed as a technical limitation in the revised version.

6. Figure 1. How did the authors calculate the a-SMA and TGFb positive cells, especially TGFb is an ECM marker.

Thanks. We have added the definition of positively stained cells in the Method section in the text (see p. 13, Line 16-18 in the manuscript):

“ImageJ (v1.52a, National Institutes of Health, USA) was used to quantify the positively stained cells, which referred to those cells stained by both blue color from DAPI and green or red signals from the secondary antibodies.”

7. How long did Dex treatment last? Was it stopped at 3 weeks or was it continued together with metformin treatment? If continued, how did they give the different treatments? Please describe those in detail. If Dex was not continued, IOP could possibly reduce in 3 weeks. Please compare if IOP in PBS group has statistically changed or not with that when Dex stropped.

Thanks. Please refer to “Essential revisions 1”.

8. Figure 2, several other control groups should also be compared, like PBS only, PBS+metformin eyedrops, and PBS+metformin oral without Dex treatment.

We appreciate your comments. We have added the control groups according to your suggestion. Please refer to “Essential revisions 1”.

9. Besides the in vitro tBHP induced oxidative cell model, it would be more convincing if the authors could also use Dex-treated cells to detect the metformin anti-oxidation effect since they used Dex induced mouse model.

Thanks for this important suggestion. Please refer to “Essential revisions 4”.

10. Figure 4, how many biological repeats and technical repeats? One of the characteristics of TM cells is that TM cells have increased expression of myocilin after exposure to Dex. Figure 4B didn't show myocilin expression increasing after Dex.

Thanks for the suggestion. We have added the related information (please refer to “Essential revisions 2”) and changed the Figure 4B to a representative one in the manuscript.

11. Fig6, it's not clear what the authors marked. How did they mark the microfilaments? A statistical analysis with graphs would be more clear and convincing.

Thanks for your comments and suggestions. We have marked representative microfilaments with larger magnified (150000×) figures embedded in Figure 6 and added statistical analysis according to your advice. The related methods and results have been shown in the manuscript (see p. 13, Line 12-16 in the manuscript, and Figure 6 in figure legend):

“For microfilament quantification, five fields with 80000× magnification of each group were captured with TEM, within which two were from the cell cortex region (defined as within 400 nm beneath the cell membrane) (*Svitkina*,*2020*) and the remaining three were beyond this area.”

12. The discussion of this study lacks further elaboration. There was no mention of the limitation of the study.

Thanks for this important suggestion. We have revised the Discussion and added the limitation as follows (see p.10, Line 11-18 in the manuscript).

“While experimental data were encouraging, there were limitations in our study. First, the concentration of MET in the anterior chamber was not measured. Additionally, we did not investigate its toxicity on ocular surfaces, such as tear film stability and components, which may change with the application of MET. Second, the sample size of the animals was relatively small, and a larger sample is needed to verify the protective effect of MET on TM. Third, the experimental period was too short to prove the long-term effects of MET and the presented figures were not in the same direction. Thus, additional studies are needed in the future.”

References:

Hartwig J, Loebel M, Steiner S, Bauer S, Karadeniz Z, Roeger C, Skurk C, Scheibenbogen C, Sotzny F. 2021. Metformin attenuates ROS via FOXO3 activation in immune cells. *Front Immunol* 12:581799. doi:10.3389/fimmu.2021.581799

Lee JY, Lee SH, Chang JW, Song JJ, Jung HH, Im GJ. 2014. Protective effect of metformin on gentamicin-induced vestibulotoxicity in rat primary cell culture. *Clin Exp Otorhinolaryngol* 7:286-294. doi:10.3342/ceo.2014.7.4.286

Li B, Zhou P, Xu K, Chen T, Jiao J, Wei H, Yang X, Xu W, Wan W, Xiao J. 2020. Metformin induces cell cycle arrest, apoptosis and autophagy through ROS/*JNK* signaling pathway in human osteosarcoma. *Int J Biol Sci* 16:74-84. doi:10.7150/ijbs.33787

Li G, Lee C, Read AT, Wang K, Ha J, Kuhn M, Navarro I, Cui J, Young K, Gorijavolu R, Sulchek T, Kopczynski C, Farsiu S, Samples J, Challa P, Ethier CR, Stamer WD. 2021. Anti-fibrotic activity of a rho-kinase inhibitor restores outflow function and intraocular pressure homeostasis. *ELife* 10doi:10.7554/*eLife*.60831

Ota S, Horigome K, Ishii T, Nakai M, Hayashi K, Kawamura T, Kishino A, Taiji M, Kimura T. 2009. Metformin suppresses glucose-6-phosphatase expression by a complex I inhibition and AMPK activation-independent mechanism. *Biochem Biophys Res Commun* 388:311-316. doi:10.1016/j.bbrc.2009.07.164

Svitkina TM. 2020. Actin cell cortex: Structure and molecular organization. *Trends Cell Biol* 30:556-565. doi:10.1016/j.tcb.2020.03.005

Warkad MS, Kim CH, Kang BG, Park SH, Jung JS, Feng JH, Inci G, Kim SC, Suh HW, Lim SS, Lee JY. 2021. Metformin-induced ROS upregulation as amplified by apigenin causes profound anticancer activity while sparing normal cells. *Sci Rep* 11:14002. doi:10.1038/s41598-021-93270-0

Yang J, Zhou Y, Xie S, Wang J, Li Z, Chen L, Mao M, Chen C, Huang A, Chen Y, Zhang X, Khan N, Wang L, Zhou J. 2021. Metformin induces ferroptosis by inhibiting UFMylation of SLC7A11 in breast cancer. *J Exp Clin Cancer Res* 40:206. doi:10.1186/s13046-021-02012-7